# Irrespective of Plaque Activity, Multiple Sclerosis Brain Periplaques Exhibit Alterations of Myelin Genes and a TGF-Beta Signature

**DOI:** 10.3390/ijms232314993

**Published:** 2022-11-30

**Authors:** Serge Nataf, Marine Guillen, Laurent Pays

**Affiliations:** 1Bank of Tissues and Cells, Hospices Civils de Lyon, Hôpital Edouard Herriot, Place d’Arsonval, F-69003 Lyon, France; 2Stem-Cell and Brain Research Institute, 18 Avenue de Doyen Lépine, F-69500 Bron, France; 3Lyon-Est School of Medicine, University Claude Bernard Lyon 1, 43 Bd du 11 Novembre 1918, F-69100 Villeurbanne, France

**Keywords:** multiple sclerosis, periplaques, myelin maintenance, demyelination, TGF-beta, MAG, NDRG1, active plaques, silent plaques, silent progression

## Abstract

In a substantial share of patients suffering from multiple sclerosis (MS), neurological functions slowly deteriorate despite a lack of radiological activity. Such a silent progression, observed in either relapsing-remitting or progressive forms of MS, is driven by mechanisms that appear to be independent from plaque activity. In this context, we previously reported that, in the spinal cord of MS patients, periplaques cover large surfaces of partial demyelination characterized notably by a transforming growth factor beta (TGF-beta) molecular signature and a decreased expression of the oligodendrocyte gene *NDRG1* (N-Myc downstream regulated 1). In the present work, we re-assessed a previously published RNA expression dataset in which brain periplaques were originally used as internal controls. When comparing the mRNA profiles obtained from brain periplaques with those derived from control normal white matter samples, we found that, irrespective of plaque activity, brain periplaques exhibited a TGF-beta molecular signature, an increased expression of *TGFB2* (transforming growth factor beta 2) and a decreased expression of the oligodendrocyte genes *NDRG1* (N-Myc downstream regulated 1) and *MAG* (myelin-associated glycoprotein). From these data obtained at the mRNA level, a survey of the human proteome allowed predicting a protein–protein interaction network linking TGFB2 to the down-regulation of both *NDRG1* and *MAG* in brain periplaques. To further elucidate the role of NDRG1 in periplaque-associated partial demyelination, we then extracted the interaction network linking NDRG1 to proteins detected in human central myelin sheaths. We observed that such a network was highly significantly enriched in RNA-binding proteins that notably included several HNRNPs (heterogeneous nuclear ribonucleoproteins) involved in the post-transcriptional regulation of *MAG*. We conclude that both brain and spinal cord periplaques host a chronic process of tissue remodeling, during which oligodendrocyte myelinating functions are altered. Our findings further suggest that TGFB2 may fuel such a process. Overall, the present work provides additional evidence that periplaque-associated partial demyelination may drive the silent progression observed in a subset of MS patients.

## 1. Introduction

Neuropathological studies classically distinguish three types of multiple sclerosis (MS) plaques: active plaques, chronic-active plaques and silent plaques. Such a classification, essentially based on the density and distribution of immune cells within plaques, has been progressively refined by the development of molecular and imaging approaches [1]. Thus, in the last years, specific attention has been given to the so-called smoldering plaques [2], a sub-category of chronic-active plaques defined by: (i) their slow expansion, measurable *in vivo* by magnetic resonance imaging (MRI) [3,4] and (ii) the abundance of iron-containing activated macrophages/microglia in theirs rims [2]. Smoldering plaques have been claimed to fuel the inexorable aggravation of neurological disability in patients with MS progressive forms [5,6]. However, such a view has been contested [7,8], leaving unanswered the question as to how disability may progress despite an apparent lack of radiological activity [9]. To address this issue, major research efforts have focused on not only plaques located in the white matter but also grey matter lesions as well as diffuse axonal loss [10,11]. In contrast, the perilesional regions, also known as periplaques, have been poorly studied so far. Indeed, periplaques are rarely viewed as a specific pathophysiological compartment and some authors even consider the term “periplaque” as synonymous to “normal-appearing white matter” (NAWM). However, while complete demyelination is by definition absent from periplaques, we previously demonstrated that periplaques are forming large areas of partial demyelination extending away from plaque borders [12]. Such large periplaques, which were observed in the spinal cord (SC) of patients with MS progressive forms, exhibited molecular hallmarks of a low-grade inflammatory process associated with subtle signs of astrocyte activation [12]. Gene co-expression analyses further demonstrated a TGF-beta molecular signature and identified *NDRG1* (N-Myc downstream regulated 1) as a hub oligodendrocyte gene targeted by the demyelination process [13]. Interestingly, although known as a tumor suppressor gene expressed by a large range of non-neural cell types [14,15], *NDRG1* exerts a key role in the maintenance of myelin sheaths [16,17]. Accordingly, in the human species, a *NDRG1* loss-of-function mutation drives the clinical manifestations of Charcot–Marie–Tooth disease type 4D [18], an autosomal recessive disorder targeting the peripheral myelin and, in some cases, the subcortical white matter [19]. Importantly, mice with a conditional knock-out of *Ndrg1* in oligodendrocytes exhibit lower levels of brain myelin proteins and a greater susceptibility to cuprizone-induced central demyelination [20]. Finally, in MS brains, *NDRG1* was shown to be epigenetically silenced in “pathology-free” regions (i.e., non-plaque areas) [21]. However, the molecular functions exerted by NDRG1 in the oligodendrocyte lineage are not fully understood yet. We reported that a highly significant number of genes that were co-downregulated with *NDRG1* in SC periplaques are involved in the translation machinery [13]. On this basis, we hypothesized that partial demyelination in SC periplaques might be due to a TGF-beta-induced down-regulation of *NDRG1*, leading in turn to an altered translation of myelin genes [13].

Despite our initial observations, periplaques have remained a somehow neglected topic until now. To our knowledge, only one published work reported on molecular data obtained from MS periplaques [22]. Moreover, periplaques in this study were essentially used as internal controls allowing to explore plaque rims, the main focus of this research work [22]. Of note, this publicly available dataset also included the mRNA profiles obtained from control non-MS normal white matter samples. In the present work, we thus re-assessed this dataset and performed comparisons between MS periplaques and control non-MS normal white matter samples. We found that, irrespective of plaque activity, brain periplaques harbored two important features that we previously reported in SC periplaques: (i) a significant decreased expression of *NDRG1* and (ii) a TGF-beta molecular signature. Since these findings pointed again to a specific involvement of NDRG1 in periplaque-associated partial demyelination, we then mined protein interaction databanks in order to identify potential molecular pathways linking TGF-beta to NDRG1 in MS periplaques.

## 2. Results

### 2.1. Irrespective of Plaque Activity, the Myelin-Related Genes MAG, NDRG1, OLIG1 and OLIG2 Exhibit Decreased Expression Levels in Periplaques

We performed comparisons between four groups of samples defined in the dataset GSE108000 [22] as: (i) control white matter tissue (control WM samples) obtained from the brains of subjects without CNS disorder, (ii) rims of chronic active lesions, (iii) periplaques of chronic active lesions (thereafter referred to as “active periplaques”) and (iv) periplaques of silent lesions (thereafter referred to as “silent periplaques”). Three pairwise comparisons were performed in which samples from each MS tissue compartment (rims, active periplaques or silent periplaques) were compared with control WM samples. Only probes exhibiting |FC| > 1.20 (|logFC| > 0.27) or |FC| < 1.20 (|logFC| < 0.27) and adjusted *p*-value < 0.05 were retained as being differentially expressed (Appendix A).

For each pairwise comparison, we extracted data obtained for eight prototypical myelin-related genes (*MBP* (myelin basic protein), *MOBP* (myelin-associated oligodendrocyte basic protein), *PLP1* (proteolipid protein 1), *MOG* (myelin oligodendrocyte glycoprotein), *MAG* (myelin-associated glycoprotein), *NDRG1*, *OLIG1* (oligodendrocyte transcription factor 1) and *OLIG2* (oligodendrocyte transcription factor 2)*);* three astrocyte-specific genes (*GFAP* (glial fibrillary acidic protein), *GJA1* (gap junction protein alpha 1) and *AQP4* (aquaporin 4)); three microglia/macrophages-specific genes (*AIF1* (allograft inflammatory factor 1), also known as IBA1, *CD14* (CD14 molecule) and *CD68* (CD68 molecule)). As expected, in rims of chronic active lesions, seven out of the eight assessed myelin-related genes displayed significantly decreased mRNA levels, indicating a full-blown demyelination (Figure 1A). Increased mRNA levels of *GFAP* and *AQP4* on the one hand and *AIF1*, *CD68* and *CD14* on the other hand indicated the co-occurrence of astrocytosis and microgliosis, respectively (Figure 1A). In active periplaques, increased levels of *AQP4* and *CD68* mRNA levels signed the existence of a glial reaction, which was however less pronounced than in the rims of chronic active lesions (Figure 1B). In the same way, fewer myelin-related genes exhibited decreased expression levels in active periplaques. Such genes comprised *MAG*, *MBP*, *NDRG1*, *OLIG1* and *OLIG2* (Figure 1B). Surprisingly, in silent periplaques, we also found that *MAG*, *NDRG1*, *OLIG1* and *OLIG2* displayed decreased mRNA levels, while astrocyte- and microglia/macrophages-related genes remained unchanged compared with control WM samples (Figure 1C). These findings indicate that myelin gene alterations extend beyond plaque borders and persist over time, irrespective of plaque activity.

### 2.2. TGFB2 and a Specific Set of Genes Coding for Cytokines/Chemokines or Growth Factors Are Over-Expressed in Periplaques

To assess the cytokine milieu in rims and periplaques, we then extracted data for a list of 193 genes of interest coding for cytokines/chemokines and growth factors (Appendix A). As expected, in the rims of chronic active plaques, multiple cytokine/chemokine genes displayed increased mRNA levels. These notably include the cytokine genes *TGFB2* (transforming growth factor beta 2), *IL1A* (interleukin-1 alpha), *IL15* (interleukine-15), *IL16* (interleukine-16), *IL18* (interleukine-18) and *IL33* (interleukine-33) as well as the matrix metalloprotease *MMP2* (matrix metallopeptidase 2),the chemokine genes *CCL2* (C-C motif chemokine ligand 2), *CCL4* (C-C motif chemokine ligand 4), *CCL18* (C-C motif chemokine ligand 18) and *CXCL16* (C-X-C motif chemokine ligand 16) and the granzyme-coding genes *GZMH* (granzyme H) and *GZMK* (granzyme K) (Figure 2A, right side on the *x*-axis).

In active periplaques, fewer cytokine/chemokine genes exhibited increased expression levels; these comprise *TGFB2, IL1A*, *IL15*, *IL33*, *CCL2* and *MMP2* (Figure 2B, right side on the *x*-axis). Finally, in silent periplaques, *TGFB2, IL15* and *CCL18* were the only cytokine/chemokine genes exhibiting increased expression levels (Figure 2C, right side on the *x*-axis). Of note, besides cytokine/chemokine genes, several genes coding for growth factors harbored increased mRNA levels in rims and/or periplaques (Figure 2A–C, left side on the *x*-axis). Overall, only five genes coding for cytokines/chemokines or growth factors were up-regulated in periplaques, irrespective of plaque activity: *TGFB2*, *IL15*, *ANGPT1* (angiopoietin 1), *HGF* (hepatocyte growth factor) and *EGF* (epidermal growth factor). On this basis, we searched for inverse correlation links between the mRNA levels displayed by such up-regulated genes and those of myelin-related genes that are down-regulated in both active and silent periplaques (i.e., *MAG*, *NDRG1*, *OLIG1* and *OLIG2*). We found that the mRNA levels of each of these oligodendrocyte-specific genes were significantly inversely correlated with those of *TGFB2*, *EGF* and *HGF* (Table 1). In contrast, such inverse correlations were not observed for *IL15* and *ANGPT1*. Such data left open the possibility that TGFB2, EGF and/or HGF might be responsible, at least in part, for the down-regulation of oligodendrocyte-specific genes in periplaques, irrespective of plaque activity.

### 2.3. Irrespective of Plaque Activity, a Significant Number of Genes Upregulated in Periplaques Are Involved in the TGF-Beta Receptor Signaling Pathway

We then attempted to determine whether genes involved in the signaling pathways of *TGFB2*, *IL15*, *ANGPT1*, *HGF* and/or *EGF* were significantly increased in active and/or silent periplaques. Using the NIH-run integrated library of pathways “Bioplanet” [23], enrichment analyses were performed on the lists of genes that were found to be significantly up-regulated in periplaques. It should be noticed that the IL-15 receptor signaling pathway is not listed in the Bioplanet library and consequently was not assessed here. The adjusted *p*-values obtained for the following signaling pathways were extracted:Signaling by TGF-beta receptor complex;EGF/EGFR signaling pathway;Signaling events mediated by hepatocyte growth factor receptor (c-Met);Angiopoietin receptor Tie2-mediated signaling.

Irrespective of plaque activity, genes up-regulated in periplaques were found to be significantly enriched in genes belonging to the “signaling by TGF-beta receptor complex” pathway (Table 2). Such an enrichment was also observed in the list of genes up-regulated in the rims of chronic active plaques (Table 2). Significant enrichments were inconstantly observed for the other assessed pathways (Table 2). Interestingly, the highest and most significant number of genes belonging to the “signaling by TGF-beta receptor complex” pathway was observed among genes up-regulated in silent periplaques (n = 24, *p*-value = 9.5 × 10^−4^). A deeper analysis was thus performed on this set of genes.

### 2.4. Silent Periplaques Exhibit a Genomic Signature Linking TGFB2 to the NDRG1-Silencing Molecule MYCN

*NDRG1* has been initially identified and named on the basis of experiments demonstrating that MYCN represses the transcription of *NDRG1* [24]. Since this stemming work, MYCN (N-Myc proto-oncogene protein) was shown to repress the transcription of multiple genes and to do so via mechanisms involving its physical interaction with both the transcription factor SP1 (transcription factor Sp1) and the histone deacetylases HDAC1 (histone deacetylase 1), HDAC2 (histone deacetylase 2) and/or HDAC3 (histone deacetylase 3) [25]. More specifically, MYCN-SP1 dimers were found to bind the GC-rich promoter regions of target genes and to induce transcriptional repression via the recruitment of HDAC1, HDAC2 and/or HDAC3 [25]. Of note, SP1 belongs to the TGF-beta receptor signaling pathway and the *NDRG1* promoter exhibits a functional GC-rich SP1-binding region [26]. Based on a survey of the human proteome, we previously suggested that, in myelinating oligodendrocytes, an engagement of the TGF-beta receptors might trigger a SP1/MYCN-mediated repression of *NDRG1* [13]. In the present work, we report that *NDRG1* down-regulation in silent periplaques is paralleled by a significant up-regulation of both *SP1* and *HDAC2*. Moreover, the 24 TGF-beta pathway genes that are up-regulated in silent periplaques comprise not only *SP1* but five genes coding for SP1 protein partners, namely *SMAD2* (SMAD family member 2), *SMAD4* (SMAD family member 4), *PARP1* (poly(ADP-ribose) polymerase 1), *NCOR1* (nuclear receptor corepressor 1) and *RBL1* (RB transcriptional corepressor like 1) (Appendix A). Such a set of 24 TGF-beta pathway genes is thus highly significantly enriched in genes coding for SP1 protein partners (enrichment factor: 15.6, *p*-value: 2.18 × 10^-6^) (Appendix A). Finally, while SP1 serves as a functional endpoint for a multitude of signaling pathways, we observed that the list of currently known SP1 protein partners is enriched in molecules of the “TGF-beta signaling pathway” (*p*-value: 2.52 × 10^-9^) (Appendix A). Since myelinating oligodendrocytes express SP1 [27], MYCN [28], NDRG1 [20] as well as TGF-beta receptors [29,30], our data indicate that, in silent periplaques, a molecular pathway linking TGFB2 to the repression of *NDRG1* may take place in mature oligodendrocytes.

### 2.5. NDRG1 Interacts with mRNA-Binding Proteins Involved in the Post-Transcriptional Regulation of Myelin Genes

We sought to obtain insights into the molecular mechanisms that might link *NDRG1* down-regulation to an alteration of myelin integrity in MS periplaques. To this aim, we surveyed the “BioGrid” protein–protein interaction database [31] and extracted the currently known list of experimentally demonstrated NDRG1 protein partners (Appendix A). Taking advantage of a recent study mapping the human CNS myelin proteome [32], we then crossed the NDRG1 interactome with the human myelin proteome (Appendix A). Interestingly, the list of NDRG1 protein partners was found to be highly significantly enriched in human myelin proteins (enrichment factor 6.24; *p*-value: 1.59 × 10^−40^) (Appendix A). It is all the more noteworthy that the NDRG1 interactome that was extracted from the BioGrid database did not comprise any interaction identified in neural cells, whether these were myelinating or not (Appendix A). Surveying the BioGrid database allowed us to demonstrate that the myelinic NDRG1 partners form a dense interaction network (Figure 3) in which the main hubs (defined by proteins exhibiting more than 20 protein interactions) comprise the two RNA-binding proteins HNRNPU (heterogeneous nuclear ribonucleoprotein U) and HNRNPH1 (heterogeneous nuclear ribonucleoprotein H1).

## 3. Discussion

Confirming our previous results on spinal cord periplaques [12,13,33], the present work provides further evidence that periplaques cannot be considered as a disease-free tissue. In patients with a progressive form of MS, periplaques appear to be characterized by the constant presence of subtle myelin alterations accompanied by a TGF-beta molecular signature. The current study refines our previous findings as it takes into account the impact of plaque activity on the periplaque molecular profile. Strikingly, even in the absence of plaque-associated inflammation, myelin alterations and a persisting process of tissue remodeling were demonstrated in silent periplaques. This observation suggests that large areas of so-called NAWM may indeed correspond to periplaque regions wherein significant myelin alterations persist. Interestingly, in contrast with active periplaques, silent periplaques did not exhibit molecular hallmarks of astrocytosis or microgliosis. The increased mRNA levels of cytokines/chemokines and growth factors observed in silent periplaques appears thus to persist in the absence of patent gliosis. As a consequence, such a process is likely undetectable under classical neuropathological examination. This holds true also with regard to periplaque-associated myelin alterations. In fact, in a large majority of neuropathological studies performed on MS brains or spinal cords, the integrity of myelin is assessed by Luxol fast blue colorations and/or PLP1 immunostainings. Such approaches do not take into account the possibility that subtle and complex modifications of myelin composition may occur.

Irrespective of plaque activity, *TGFB2* was up-regulated in periplaques along with a highly significant number of genes involved in the TGF-beta signaling pathway. This was not the case for any of the more than 190 cytokine/chemokine or growth factor genes and associated pathways that we assessed in parallel. Such findings confirm our previous results and indicate that one or several cell types in periplaques engage a TGF-beta-driven genomic program. Cells of the oligodendrocyte lineage express the TGF-beta receptors [29,30] and are thus potentially targeted by TGFB2. A survey of the protein interaction database “Biogrid” allowed us to build up a putative molecular pathway linking TGF-beta receptors to SP1 and HDAC2 (Figure 4), two MYCN partners involved in the MYCN-induced down-regulation of *NDRG1* [25,26]. Genes coding for the key molecules of such a putative pathway are up-regulated in silent periplaques (Figure 4) and their expression at the protein level were previously demonstrated in myelinating oligodendrocytes. Besides NDRG1 and the TGF-beta receptors, this is notably the case for SP1 [27], HDAC2 [34], MYCN [28], SMAD2 [35] and SMAD4 [36]. Experimental approaches are needed to determine whether TGFB2 may actually induce the down-regulation of *NDRG1* in mature oligodendrocytes.

Another important point of our results relates with the potential mechanisms linking *NDRG1* down-regulation to the process of periplaque-associated partial demyelination. Only a few studies have attempted to obtain molecular insights into the functions of *NDRG1* in CNS demyelinating disorders [20,21]. Based on a survey of proteomics datasets, we found that NDRG1 and a highly significant number of NDRG1 protein partners exhibit abundant levels in human central myelin sheaths. Moreover, RNA-binding proteins were significantly over-represented among such NDRG1 myelinic partners. These findings indicate that NDRG1 is possibly involved in the now well-described post-transcriptional mechanisms regulating the expression of myelin genes such as *MAG* [37,38,39], *MBP* [40,41,42] and *MOBP* [43,44]. In particular, *MAG* was demonstrated to be post-transcriptionally regulated by a complex of splicing proteins comprising notably HNRNPA1 (heterogeneous nuclear ribonucleoprotein A1) and the myelinic NDRG1 protein partner HNRNPH1 [45]. This point may be of particular interest, since *MAG* was the only myelin gene exhibiting decreased mRNA levels in silent periplaques. In the myelinating oligodendrocytes of silent periplaques, a molecular pathway linking TGFB2 to the repression of *NDRG1* and a subsequent decreased expression of *MAG* may thus operate (Figure 4). More generally, the down-regulation of *NDRG1* in oligodendrocytes might alter the amounts of specific myelin proteins while sparing other myelin proteins. In support of this view, reports published in the 1980s by the team of Wallace Tourtelotte showed that MS periplaques are characterized by a process of partial demyelination that was more pronounced for MAG than for MBP or PLP1 [46,47]. Notably, in the outer periplaque regions, MAG was the only myelin protein exhibiting decreased protein levels compared with adjacent NAWM areas [46,47].

Irrespective of plaque activity, *TGFB2,* but not *TGFB1,* was up-regulated in brain periplaques. This observation is in contrast with our previous reports that, based on gene co-expression analyses, pointed to the potential role of *TGFB1* rather than *TGFB2* in spinal cord periplaques [13,33]. Such a discrepancy might be due to differences regarding the analytical approaches (identification of differentially expressed genes vs. identification of co-expression networks), the regional localization of samples (spinal cord vs. brain) and/or the nature of comparisons (periplaques vs. control tissue in the present work, eriplaques vs. NAWM in our previous studies). Although sharing the same receptors and signaling pathways, TGFB1 and TGFB2 exert distinct and non-redundant functions [48,49]. In particular, *Tgfb2* KO mice do not display the disseminated autoinflammatory phenotype observed in *Tgfb1* KO mice [48,49]. Such isoform specificities are notably due to differences regarding the affinities of TGFB1 and TGFB2 for the TGF-beta receptors TGFBR1 (TGF-beta receptor type-1), TGFBR2 (TGF-beta receptor type-2) and TGFBR3 (TGF-beta receptor type-3) [50,51]. Moreover, TGF-beta isoforms exhibit distinct expression patterns [52] and the mechanisms driving the activation of latent TGF-beta molecules are isoform-specific [52]. Interestingly, a survey of the “Human Protein Atlas”, currently the largest expression database for human genes and proteins [53,54], indicates that astrocytes represent a major source of TGFB2, whereas TGFB1 mostly derives from immune cells (Appendix A). In terms of therapeutic perspectives, the inhibition of TGFB1 bears the risk of fueling neuroinflammation. In contrast, inhibiting TGFB2 may dampen astrocytosis and prevent periplaque-associated myelin alterations, while leaving unchanged any TGFB1-mediated immunoregulatory pathway. Favoring this view, in distinct murine models of TGF-beta-mediated fibrosis, a monoclonal antibody targeting specifically TGFB2 was shown to spare the immunoregulatory functions of TGFB1 while exerting potent therapeutic effects [52].

Although based on the mining of experimental data, our work obviously calls for further experiments in order to ascertain the pathophysiological scheme that we propose. In particular, it appears important to determine whether NDRG1 is involved in the post-transcriptional regulation of specific myelin genes (notably *MAG*) in myelinating adult oligodendrocytes. In mice, a conditional and inducible KO of *Ndrg1* in adult oligodendrocytes should be assessed *in vitro* and *in vivo*. Furthermore, the murine model of MS, experimental autoimmune encephalomyelitis (EAE), should be used to reach two main objectives: (i) assessing the impact of Tgfb2 (either exogenously administered or endogenously produced via genetically-modified astrocytes) on inflammation, Ndrg1 expression and demyelination and (ii) evaluating the therapeutic effects of strategies aimed at inhibiting Tgfb2 (either by the administration of an anti-Tgfb2 monoclonal antibody or via a conditional and inducible KO of *Tgfb2* in astrocytes). Similar approaches could be applied *in vitro* on human myelinating oligodendrocytes. Finally, periplaques in the brains and spinal cords of MS patients should be screened for the expression of multiple myelin proteins, including MAG, MOG, MBP and PLP1.

## 4. Materials and Methods

### 4.1. Bioinformatics Analyses of Genomics Data

Data from Hendrickx DA et al. [22] were extracted from the NIH-run databank GEO dataset (GSE10800). Normalized data are presented in Appendix A. Comparisons between groups were performed in parallel with the GEO-embedded analytical webtool GEO2R and an R package specifically dedicated to the analysis of GEO datasets [55]. Only probes with a |FC| > 1.20 (|logFC| > 0.27) or |FC| < 1.20 (|logFC| < 0.27) and an adjusted *p*-value < 0.05 were considered as differentially expressed. For genes covered by distinct probes, a mean FC was calculated from the values reported for each probe exhibiting an adjusted *p*-value < 0.05. The lists of significantly up-regulated or down-regulated genes were highly similar when comparing the 2 analytical approaches mentioned above (>99% similarity). The more conservative method (GEO2R) was chosen for further analyses on differentially expressed genes. Enrichment analyses were performed on the web platform “Enrichr” [56] and using the Bioplanet library [23], an NIH-sourced large bank of pathways integrating annotations from multiple libraries including KEGG (Kyoto encyclopedia of genes and genomes) [57], Reactome [58] and WikiPathways [59]. Bioinformatics analyses of genomics data were all repeated at least 3 times and were updated in October 2022.

### 4.2. Bioinformatics Analyses of Proteomics Data

The lists of currently known protein partners of SP1 and NDRG1 were extracted from the Biogrid database [31], which hosts ~1.93 million curated protein and genetic interactions in several species. We retained only protein–protein interaction data obtained by experimental approaches (i.e., not inferred from in silico analyses) and demonstrated in the human species. When needed, the statistical significance of enrichment factors was assessed using the Fisher’s exact test. Bioinformatic analyses of proteomics data were all repeated at least 3 times and were updated in October 2022.

## 5. Conclusions

Our data show that, in MS progressive forms, periplaques surrounding brain lesions harbor two important features that we previously identified in spinal cord periplaques: (i) a process of partial demyelination notably characterized by a down-regulation of *NDRG1* and (ii) a TGF-beta molecular signature. We further demonstrate that such alterations persist in silent periplaques and that TGFB2, rather than TGFB1, is likely the key driver of such alterations. From a therapeutic point of view, our results suggest that dampening specifically TGFB2 may spare the anti-inflammatory functions of TGFB1 while preventing the development of periplaque-associated tissue remodeling. In a substantial share of MS patients, the progression of disability appears disconnected from plaque-associated inflammation [9,60]. Our work raises the possibility that slowly evolving myelin alterations taking place essentially in periplaques might be involved in such a silent progression.

## Figures and Tables

**Figure 1 ijms-23-14993-f001:**
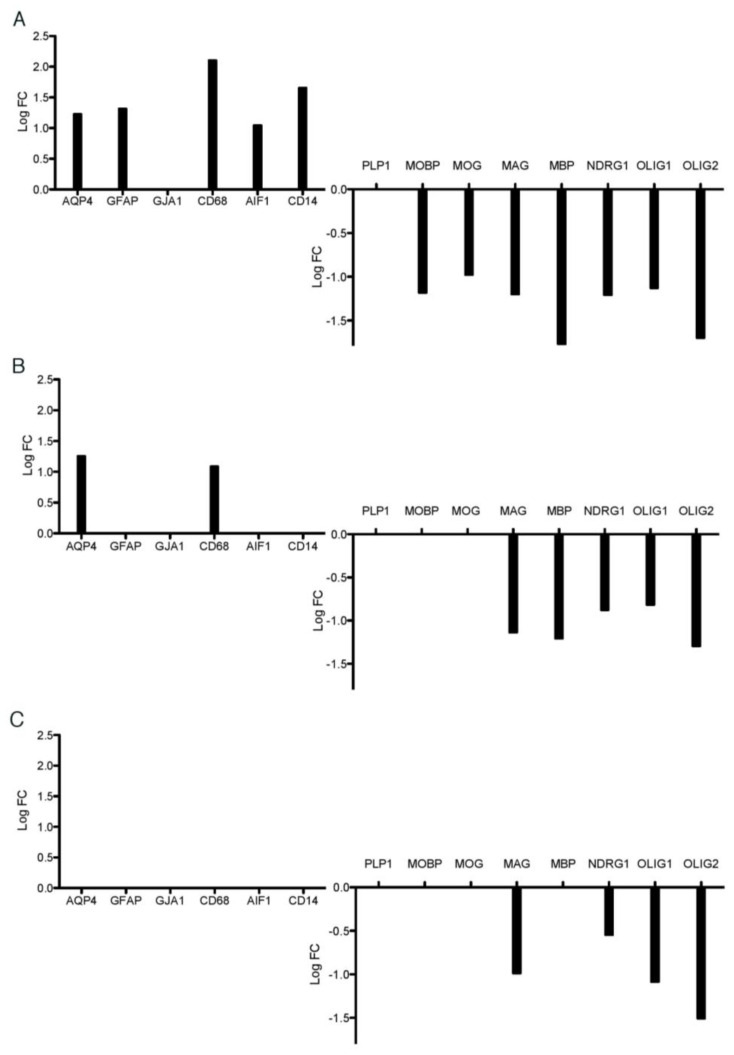
Differential expression of key astrocyte, microglial and oligodendrocyte genes in rims and periplaques compared with control white matter.

**Figure 2 ijms-23-14993-f002:**
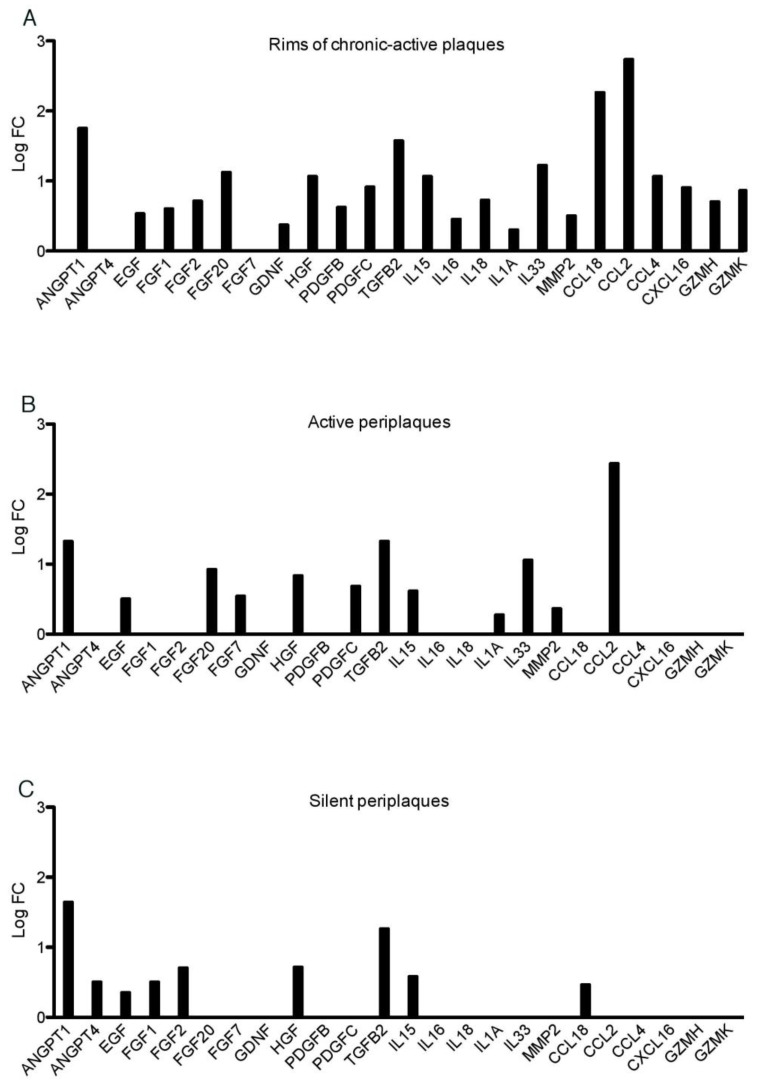
Cytokine, chemokine and growth factor genes overexpressed in rims and periplaques compared with control white matter.

**Figure 3 ijms-23-14993-f003:**
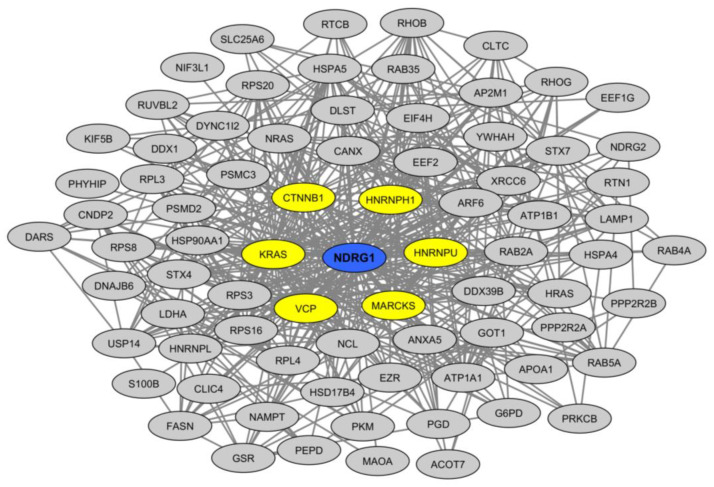
Protein–protein interaction network linking NDRG1 with myelinic protein partners. Hubs are shown in yellow.

**Figure 4 ijms-23-14993-f004:**
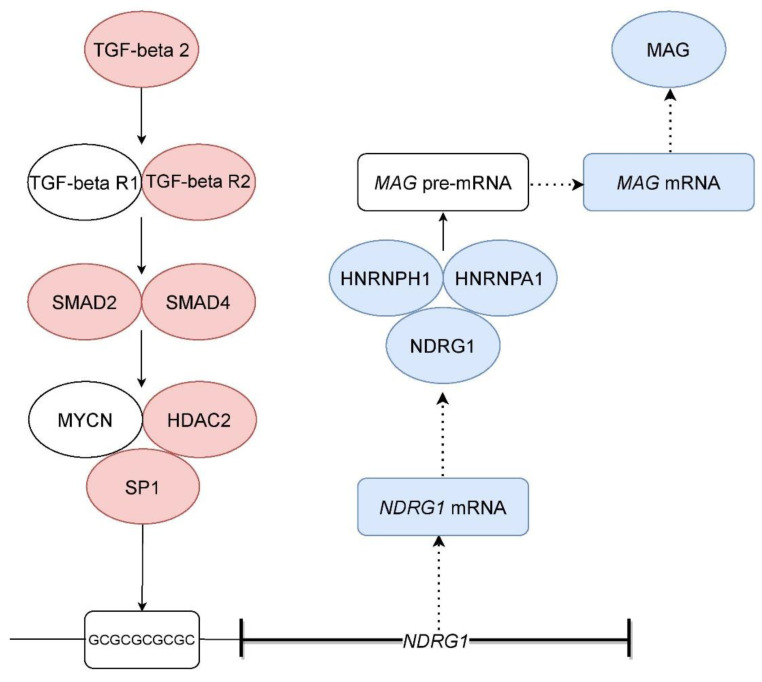
Putative molecular pathway linking TGFB2 up-regulation to the decreased expression of MAG in silent periplaques. In red: proteins encoded by genes that are up-regulated in silent periplaques. In blue: mRNAs down-regulated in silent periplaques (in italics) and proteins encoded by genes that are down-regulated in silent periplaques.

**Table 1 ijms-23-14993-t001:** Inverse correlations between the expression levels of candidate cytokine/chemokine/growth factor genes and oligodendrocyte genes.

	*MAG*	*OLIG1*	*OLIG2*	*NDRG1*
*TGFB2*	*p* = 0.0028 r = −0.58	*p* = 0.00069 r = −0.64	*p* = 0.00016 r = −0.69	*p* = 0.0042 r = −0.55
*ANGPT1*	NS	NS	*p* = 0.0039 r = −0.56	0.021 r = 0.46
*EGF*	*p* = 0.027 r = −0.44	*p* = 0.044 r = −0.40	*p* = 0.00055 r = −0.65	*p* = 0.0010 r = −0.62
*HGF*	*p* = 0.00058 r = −0.65	*p* = 0.00032 r = −0.67	*p* = 0.0000086 r = −0.77	*p* = 0.000010 r = −0.77
*IL15*	NS	NS	0.00027 r = −0.67	0.013 r = −0.49

*p*-values (*p*) and correlation coefficient (r) obtained with the Spearman correlation test are indicated. NS—not significant.

**Table 2 ijms-23-14993-t002:** Enrichment analyses of the lists of genes up-regulated in rims or periplaques.

Pathway		Localization	
	Rims of CA plaques	Active periplaques	Silent periplaques
Signaling by TGF-beta receptor complex	0.015	0.010	9.5 × 10^−4^
EGF/EGFR signaling pathway	0.0038	NS	0.0030
Signaling events mediated by hepatocyte growth factor receptor (c-Met)	NS	NS	0.012
Angiopoietin receptor Tie2-mediated signaling	NS	NS	NS

Adjusted *p*-values are indicated for each pathway and localization. NS—not significant.

## Data Availability

The data generated in this work were obtained by the re-analysis of a RNA expression dataset which was previously published [22] and rendered publicly available in the databank “GEO datasets” (reference GSE10800).

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
