# Peer review of "Irrespective of Plaque Activity, Multiple Sclerosis Brain Periplaques Exhibit Alterations of Myelin Genes and a TGF-Beta Signature"

_ijms, 2022, doi:10.3390/ijms232314993_

Round 1

Reviewer 1 Report

Minor points:

1.     In line 15, there is a typo “in in this context”.

2.     In figure 2, it would be better if authors could categorize the genes by the specific function (cytokines/chemokines or growth factors/plaque activity) in the figure, instead of putting all genes together in one panel, just to better visualize the data.

3.     In figure 2, the authors missed labeling “A, B, C”

4.     Authors missed figure legends in figure 4

5.     Keep the format of words consistent in the entire paper, use TGFB1 and TGFB2 or TGF-beta1 and TGF-beta2

6.     Missing “S” in table S6 in table S1-S8

Major points:

1.     To improve the preciseness of the data, authors should add error bars in each histogram of Figure 1 and Figure 2.

Author Response

Answer to Reviewer 1

We thank reviewer 1 for her/his positive evaluation of our work. The small typos were corrected (“In this context in line 15; missing “S” in Table 6; unified format for TGFB1 and TGFB2 in the entire manuscript). We added a detailed title to Figure 4 and labelled the panels A, B and C. We generated a new version of Figure 2 in which: i)  immune-related molecules can be found on the right side of the x-axis and ii) growth factors are localized on the left side of the x-axis. Text changes have been performed accordingly. Regarding the error bars in Figure 1 and 2, we would like to underscore that the values expressed on the y-axis are “Log fold changes” for which, by definition, error bars cannot be calculated.  We feel that presenting data as expression values with error bars rather than Log fold changes would require generating separate panels for each gene, rendering thus the whole figure too heavy and less easily readable. However, to improve the preciseness of data, we added a supplementary data (Table S9) in which the normalized expression values obtained for 29833 probes were extracted from the GEO dataset GSE108000.

We hope that our paper is now suitable for publication in the International Journal of Molecular Sciences.

Reviewer 2 Report

This manuscript utilizes publicly available transcriptomic data to determine if periplaque areas are molecularly altered in MS patients.  This study provides evidence that demyelination is occurring in the perivasicular areas of lesions is associated with increased TGF-beta signaling.  The bio-informatic approaches used in this study are solid but there are no new experimental data presented to fully demonstrate that increased TGF-Beta signaling can be responsible for the reduced oligodendrocyte function.

Experimental plans to test this hypothesis should be discussed.

Thorough proofreading is needed to catch grammatical errors and figure 4 does not have a title. 

Author Response

Answer to Reviewer 2

We thank reviewer 2 for her/his positive evaluation of our work. We did our best to correct typos and grammatical errors. We added a detailed title to Figure 4. As requested by reviewer 2 we added a paragraph in the discussion section so to expose experimental plans allowing to test our hypothesis (lines 340 to 354).

We hope that our paper is now suitable for publication in the International Journal of Molecular Sciences.